# Integration of wearable devices and artificial intelligence in Alzheimer's disease: A scoping review protocol

Wenhao Qi[1,2], Yankai Shi[1], Shiying Shen[1,2], Bingsheng Wang[1], Shuai Zhang[3], Junling Kang[4], Xiaodong Lu[5], Guowei Jiang[6], Lizzy M. M. Boots[6], Qian Xu[7,8]*, Shihua Cao[1,2]

1 School of Nursing, Hangzhou Normal University, Hangzhou, China, 2 Key Laboratory of Cognitive Disorder Assessment Technology, Zhejiang Province, China, 3 Emergency and Critical Care Center, Intensive Care Unit, Zhejiang Provincial People's Hospital, Hangzhou, China, 4 Department of Neurology, The Third Affiliated Hospital of Zhejiang Chinese Medical University, Hangzhou, China, 5 Department of Neurology, Affiliated Hospital of Hangzhou Normal University, Hangzhou, China, 6 Department of Psychiatry and Neuropsychology and Alzheimer Center Limburg, School for Mental Health and Neuroscience Maastricht University, Maastricht, The Netherlands, 7 School of Public Health, Angeles University Foundation, Philippines, 8 Department of Nursing, Qiandongnan National Vocational and Technical College, China

* csh@hznu.edu.cn

## Abstract

The incidence of Alzheimer's disease (AD) continues to rise, and predictive models combining artificial intelligence (AI) with wearable devices offer a new approach for its detection and diagnosis. Existing reviews remain focused on traditional biomarkers, making it necessary to supplement the evidence in this field, particularly given the rapid advancements in wearable and AI technologies. The scoping review protocol aims to systematically evaluate AI-based predictive models using wearable devices for AD, with a focus on their measurement outcomes and model development processes. The review will follow the Arksey and O'Malley framework and incorporate PRISMA-ScR guidelines. This study will search multiple databases, including Web of Science, Cochrane Library, and PubMed, covering relevant gray literature. The quality of the included studies will be rigorously assessed using the Prediction Model Risk of Bias Assessment Tool (PROBAST) and Transparent Reporting of a Multivariable Prediction Model for Individual Prognosis or Diagnosis (TRIPOD) checklists. Two independent reviewers will conduct title and abstract screening, retrieve and assess full-text evidence sources, and extract data. The results will be narratively synthesized and presented in tables and figures. The knowledge gained from this review is expected to provide systematic evidence supporting AI-based predictive models that combine wearable devices for AD, potentially offering insights into model construction details such as data collection and external validation.

**Data availability statement:** No datasets were generated or analysed during the current study. All relevant data from this study will be made available upon study completion.

**Funding:** Zhejiang Province Traditional Chinese Medicine, Science, and Technology Project (2023ZF134) supported by SC, Zhejiang Provincial Medical and Health Science and Technology Program Project (2022KY1052) supported by SC,First-Class Course of Zhejiang Province (2022-1133) supported by SC,Basic Public Welfare Research Project/Joint Fund Project of Zhejiang Province: Research on accurate localization of gait disorder brain region in Parkinson's disease based on pfMRI data set and hierarchical Bayesian model(L-BY23H200002), supported by XL.

**Competing interests:** The authors have declared that no competing interests exist.

## Introduction

As the global population continues to age, the incidence and severity of dementia are increasing, posing a significant challenge to global health [1]. According to the World Health Organization, over 55 million people are affected by dementia [2], with Alzheimer's disease (AD) accounting for 60% to 70% of cases [3]. AD is primarily characterized by cognitive impairment, memory loss, and a range of motor function deficits [4]. Due to its complex etiology and mechanisms, effective treatments capable of fully reversing the disease course remain unavailable, placing a substantial burden on both society and families [5]. Therefore, early diagnosis and prediction are crucial for optimizing care management strategies and delaying disease progression. Currently, clinical diagnosis of AD mainly relies on neuropsychological assessments and biomarker testing based on the ATN framework [6]. However, these methods are costly, invasive, and complex to administer, and are prone to biases from patient self-reports and external factors [7]. With advancements in technology, wearable devices have emerged as a promising tool for early diagnosis. These devices offer non-invasive, high-frequency monitoring capabilities and provide new insights for more convenient health management, potentially serving as a valuable complement to existing diagnostic methods [8].

With the rapid advancement of AI technology, machine learning has become a critical tool in the healthcare field, particularly in disease prediction and diagnosis [9]. The fundamental principles of machine learning can be divided into three major categories: supervised learning, unsupervised learning, and reinforcement learning [10]. In AD prediction, supervised learning methods are particularly widely used due to their ability to train models based on labeled data, enabling the differentiation of various disease states. For example, algorithms such as Support Vector Machines and Random Forests have achieved significant results in tasks such as gait analysis and EEG signal analysis. Unsupervised learning also plays a crucial role in the absence of labeled data, particularly in exploring differences and characteristics of AD subtypes. Clustering algorithms can help researchers uncover potential disease patterns. Reinforcement learning, on the other hand, shows promise in personalized treatment and real-time prediction by optimizing decision-making processes, offering new directions for patient health management.. In the prediction of AD, wearable devices can provide various types of data (such as physiological signals, gait, and activity levels), and when integrated with machine learning, they enable high-frequency monitoring and personalized diagnosis [10]. Clinical prediction models use multiple variables to estimate current diagnostic outcomes (diagnostic models) or predict future disease outcomes (prognostic models). Several studies have developed predictive models for AD based on wearable devices. For example, Younghoon Jeon used SHIMMER wearable devices to monitor gait data and combined algorithms such as random forests and AdaBoost, achieving an accuracy rate of 72% [11]. Ahsan Shahzad developed a support vector machine model, achieving an accuracy rate of 70% [12]. Kyeonggu Lee utilized wearable EEG signals to build a multilayer perceptron model, achieving an accuracy rate of 74% [13]. In our recent study, we integrated multiple wearable sensors within a three-stage federated learning framework to develop a

home-based AD detection system [14]. Additionally, we constructed the EmoMarker system using wearable microphones and depth cameras, achieving a classification accuracy of 81.13% by leveraging motion and acoustic data [15].In the framework where AI is integrated with wearable devices, the selection and training of models are key to successful application. Typically, choosing an appropriate machine learning algorithm (such as deep learning, random forests, or support vector machines) and training it based on the characteristics of patient data and clinical needs can optimize the model's predictive performance. For example, when the model needs to handle complex time-series data (such as gait studies in neurodegenerative diseases), methods like recurrent neural networks and long short-term memory networks have demonstrated superior performance [16]. Similarly, graph neural networks play a crucial role in the spatiotemporal processing of EEG data by simultaneously considering the dynamic changes in the time series and the spatial relationships between different brain regions [17,18].

Additionally, despite the initial progress in applying machine learning techniques to AD prediction, several challenges remain. First, early symptoms of AD are often difficult to detect, and physiological changes are typically slow, requiring machine learning models to have high sensitivity in identifying subtle changes from low signal-to-noise ratio data. Furthermore, data gaps, individual differences, and the heterogeneity of multi-source data pose significant challenges to model training and validation. Therefore, selecting appropriate features, handling missing data, and optimizing model structures have become key factors in improving prediction accuracy and robustness.

The potential of these technologies is currently being explored. However, many reviews tend to focus on AI models for traditional biomarkers in AD, failing to address the developmental potential of wearable devices in Alzheimer's. Only two systematic reviews [19,20] have integrated wearable devices with AI models in the context of AD. Furthermore, these reviews do not adequately demonstrate the feasibility of integrating wearable devices with AI technologies. In their descriptions of wearable devices, they fail to provide detailed information on data collection processes, such as device selection, specific collection strategies and paradigms, as well as the dropout and data loss rates during the collection of different types of data (e.g., gait, eye movement, neurophysiological data), and the reasons for these issues (e.g., device malfunction, participant dropout, or non-standard data collection).

Regarding AI models, these reviews lack specific details on model construction, such as feature selection strategies, handling of missing data, sample balancing strategies, comparisons of different algorithm performances, and validation methods. These details are crucial for building wearable device-based machine learning models in Alzheimer's disease.

To fill these gaps, a comprehensive review of existing literature is needed. This review should provide a detailed analysis of the workflow from data collection to external validation of models, offering a comprehensive overview of existing model construction. It will help to understand the current state of research, analyze mainstream methods, and evaluate key improvements made so far. Specific topics for analysis and discussion include the differences in wearable device types and data collection protocols, key digital biomarkers for different types of data, whether missing data handling and sample balancing are commonly performed in current research, and the performance differences between traditional algorithms, deep learning models, and complex ensemble models. Additionally, the status of external validation in existing research should be examined in more detail. The AI models constructed in each study will be evaluated using PROBAST to assess their risk of bias and applicability.

These findings will provide valuable insights for future work, including whether systematic reviews and meta-analyses should be conducted, the development of new models, or the validation of the applicability of existing models. Given the breadth of this goal, a scoping review is the most appropriate approach [21].

The unique contribution of this protocol lies in its development of a systematic evaluation and comprehensive review of AI prediction models based on wearable devices. By providing a comprehensive review of existing research, it addresses the gap in the current literature concerning the integration of these two technologies. Specifically, we aim to report on several key dimensions, including the selection of wearable devices, data collection paradigms, feature selection strategies, model construction and training, model validation, and model transparency. In addition to analyzing the applications of

various wearable devices, we explore how advanced AI techniques can enhance their diagnostic accuracy and monitoring capabilities. This approach offers new perspectives for the early diagnosis of AD, prediction of disease progression, and personalized healthcare. The construction methods for AI-based wearable device predictive models are shown in Fig 1.

## The research on wearable device-based AI models is as follows

**Review scope questions.** This scope review aims to summarize predictive models for AD based on wearable devices and artificial intelligence algorithms. As our understanding of the literature evolves, we will adopt an iterative approach and adjust the research questions as necessary. The review will focus on the following research questions:

1. What wearable devices have been used in AD prediction models, and what types of data were collected?

2. What algorithms have been used in the literature to construct various AD prediction models?

3. What is the accuracy of AI models in predicting AD? How do multimodal models compare to unimodal models?

4. What are the limitations or shortcomings of existing wearable device-based AD prediction models?

**Contributions of this study.**

1. This review systematically compiles AI models related to wearable devices for AD, summarizing the core technologies, methods, and advancements in current research. This overview will help both the academic and clinical communities gain a comprehensive understanding of the current state of research in this field.

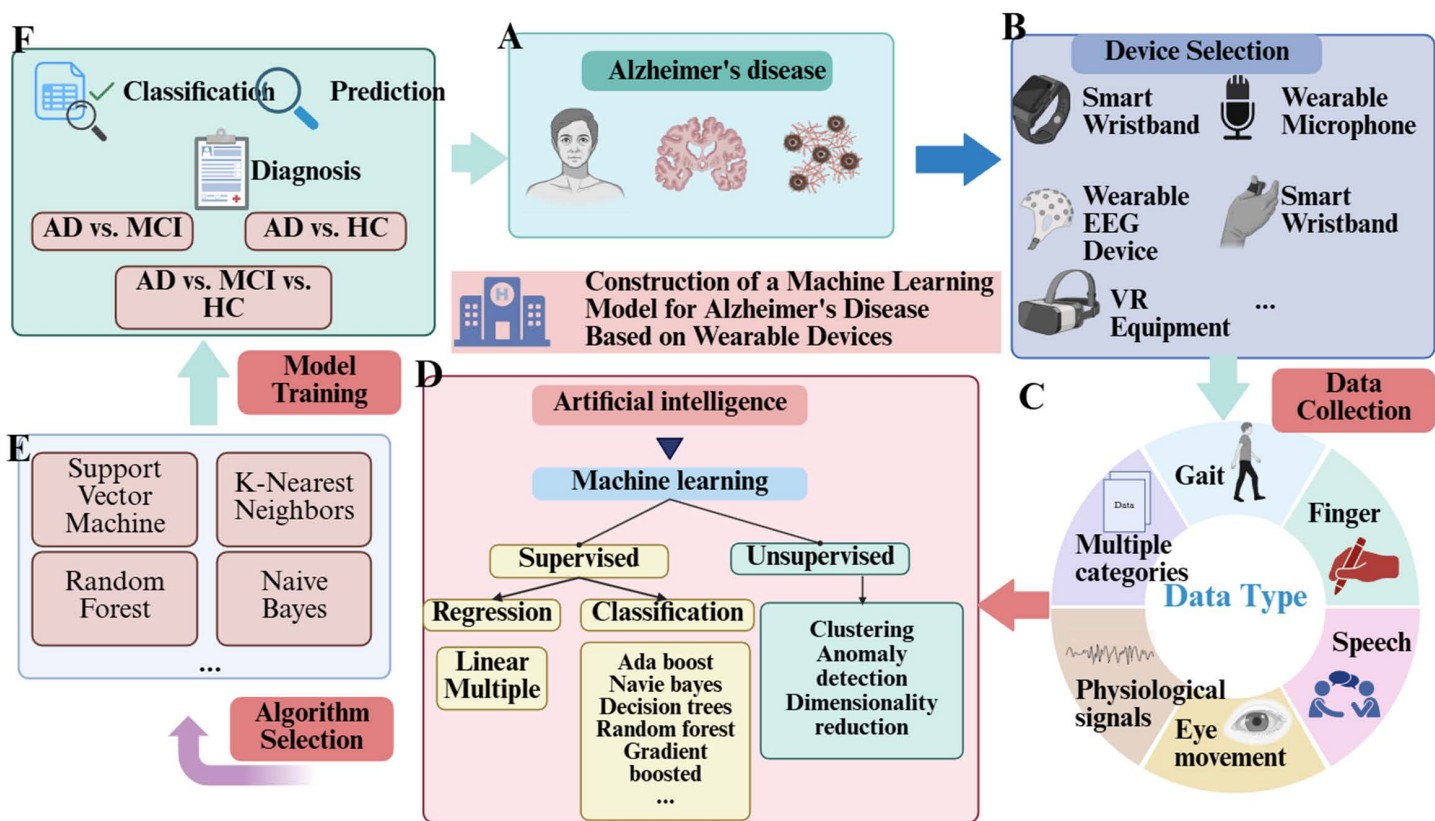

**Fig 1. Establishment methods.**

2.  By analyzing existing studies, this review identifies key gaps and limitations in the field, providing direction for future research and promoting the innovative development of AI models in AD. It also offers guidance for optimizing and improving models to better meet clinical needs.

3.  This review aims to provide new insights into the detection, diagnosis, and prediction of AD, ultimately improving diagnostic accuracy and the quality of clinical management for AD patients.

## Methods and analysis

### Registration and protocol

This protocol has been registered on the Open Science Framework (OSF), with the registration number available at: https://osf.io/nvf3x and DOI: https://doi.org/10.17605/OSF.IO/NVF3X. If necessary, we will update the protocol within the OSF registration.

### Ethics and dissemination

This scoping review solely summarizes existing published data, so ethical approval is not required. The study findings will be disseminated through peer-reviewed journals and academic conferences, focusing on the application of AI and wearable devices in Alzheimer's disease predictive models. We hope the results will provide valuable insights into the detection and diagnosis of AD and attract attention from interdisciplinary scholars. This study aims to offer useful information to researchers from diverse fields such as artificial intelligence, neuroscience, and engineering.

### Study design

This study will use a scoping review approach to outline the application of artificial intelligence algorithms in the development of wearable device-based predictive models for AD.This includes methods related to data collection (sourced from digital health devices), algorithm selection, feature selection, prediction timeframes, model quality and robustness, handling of missing data, and the integration of multimodal data. The scoping review will be developed based on the Arksey and O'Malley framework [22], incorporating the latest enhancements from the Joanna Briggs Institute [23]. This protocol follows the PRISMA-ScR reporting standards for scoping reviews, as adapted from the Preferred Reporting Items for Systematic Reviews and Meta-Analyses Extension [24] (see Supplementary S1 Table 1 online). The included studies will undergo rigorous evaluation using the Prediction Model Risk of Bias Assessment Tool and the Transparent Reporting of a Multivariable Prediction Model for Individual Prognosis or Diagnosis (TRIPOD) checklist [25,26]. The PROBAST tool is used to assess the risk of bias in prediction models, helping to evaluate whether there is bias in the design, data processing, and analytical methods of the prediction models in research. The TRIPOD tool is used to assess and ensure the completeness of the reporting for multivariable prediction models. It provides a standardized reporting guideline that helps researchers clearly and transparently report the construction, validation process, and results of prediction models. The combined use of PROBAST and TRIPOD offers a more comprehensive framework to ensure the quality and transparency of predictive model research.

### Inclusion criteria

**Participants.**  This review will focus on examining the combined use of wearable devices and artificial intelligence in AD research. Studies related to other neurodegenerative diseases (such as Parkinson's disease, vascular dementia, Lewy body dementia, etc.) will be excluded. There will be no restrictions on the specific severity of Alzheimer's disease or on subtypes, such as Late-Onset AD and Early-Onset AD [27]. No restrictions will be placed on sample size, provided that patients are diagnosed with or predicted to have AD. Patients of all demographic characteristics are eligible.

**Concept.** This study focuses on predictive models for AD built using wearable devices and artificial intelligence algorithms. This review will include all artificial intelligence algorithms, including machine learning and deep learning, but will exclude traditional prediction models such as Cox regression and AI models that do not report performance explicitly. Studies that do not report comparable predictive performance metrics will be excluded. Similarly, models based solely on traditional biomarkers (such as MRI, PET, or cerebrospinal fluid) constructed in clinical settings, or studies that validate existing models without introducing new samples, will also be excluded.

Given the wearable and portable nature of wearable devices, this review will cover everything from simple wearable sensors to complex commercial wearable devices. The application of embedded sensors in home environments and devices such as electronic road systems are not within the scope of wearable devices covered in this review [28,29]. The types and combinations of predictive variables collected by wearable devices will be reviewed (e.g., multimodal models integrating omics data with sensor data). This review will include any outcomes or cumulative results predicted by the models, as well as any time spans involved in the predictions. Studies that focus only on a single risk factor and do not present a model, or describe models that are not explicitly applicable to AD patients, will be excluded.

### Context

This review will include literature on the application of wearable device-based AI models for AD across all environments, including clinical settings (such as hospitals, laboratories, and long-term care facilities) and non-clinical settings (such as home, outdoor, community, and private residences), regardless of country, cultural, or racial background.

### Types of studies

In research on AI predictive models, various study types are included, such as retrospective studies, prospective studies, experimental designs, cross-validation and iterative studies, case studies, comparative studies, big data analyses, multi-modal data prediction studies, and causal inference research. These study types employ different methodologies and data sources to explore and optimize the performance, reliability, and practical application of predictive models. Regarding data sources, publicly available datasets or self-collected data can be used, and algorithmic architectures may be based on publicly available frameworks or independently designed architectures.

### Search strategy

This study will search the following electronic bibliographic databases: Cochrane Library, PubMed, Embase, Scopus, CINAHL, IEEE Xplore, EBSCO, PsycINFO, and Google Scholar. "Gray literature" refers to documents that have not been formally published or peer-reviewed. These materials are often difficult to retrieve through traditional academic databases but still contain valuable research findings or information. We will retrieve available research from conference papers and proceedings, preprints, and dissertations through the Opengrey database. The specific search process is as follows:

1. All team members have received professional training based on the textbook *Medical Literature Information Retrieval* [30].

2. Given the interdisciplinary nature of AI models and wearable devices, the research team comprises members with backgrounds in computer informatics, neuroscience, medical informatics, and nursing. After consulting with members of the Chinese Digital Medicine Society and collaborating with a librarian, we initially formulated search terms. The search strategy will be piloted to test the relevance of keywords and databases, and a finalized search strategy will be developed. Search strings for five of the databases are available in Supplementary S2 Table online. The search timeframe will cover the period from the inception of each database until January 30, 2025.The estimated completion time for organizing the literature is February 20, 2025.

3. Based on the established search strategy, a formal search will be conducted across all databases, and all references will be downloaded.

4. We will screen the reference lists of all identified studies to locate additional relevant literature.

## Study/source of evidence selection

After data collection, all references will be imported into EndNote X9 for management, and duplicate records from the database will be removed. Two reviewers will independently assess the titles and abstracts to evaluate the inclusion and exclusion criteria and select the original articles. The full texts of the selected studies will be reviewed in detail by the two reviewers according to the study criteria. Reasons for excluding full-text articles will be reported in the scoping review. Any disagreements between the reviewers during the selection process will be resolved through group discussion. The results of the retrieval and study inclusion process will be fully reported in the final scoping review and presented in the PRISMA-ScR flowchart [24].

Specifically, the study selection process will be divided into three steps:

Step 1: The researchers will perform an initial screening of 50 studies using EndNote X9 reference management software by reviewing titles and abstracts. The Kappa coefficient between the reviewers will be calculated. If the Kappa value is below 0.8, the search strategy will be adjusted or the screening personnel will be replaced.

Step 2: Researchers conduct the initial formal screening based on the abstract and title. In cases of disagreement between the two primary reviewers (WQ and CD), a third reviewer (SC) makes the final decision.

Step 3: The researchers will perform full-text reviews to further screen the studies. If disputes occur, a third reviewer (SC) will again make the final judgment. The number of studies excluded and the reasons for exclusion will be presented in the flowchart. This process will follow the PRISMA-ScR standards.

Accordingly, we have developed specific inclusion and exclusion criteria, along with a preliminary inclusion and exclusion flowchart based on PRISMA-ScR (Fig 2). As the study progresses, we will update this inclusion and exclusion process and the accompanying figure as needed.

Inclusion Criteria:

• The study must involve the use of wearable devices.

• The study must be written in English.

• No restrictions on the type of predictive model (including diagnostic and prognostic models).

• Predictive models constructed using machine learning or deep learning algorithms.

Exclusion Criteria:

• Studies for which the full text is unavailable or incomplete.

• Duplicate publications.

• Books, editorial materials, reviews, and other non-research type studies.

• Studies not focused on AD, instead emphasizing other dementia subtypes or mild cognitive impairment (MCI).

• Predictive models based solely on traditional biomarkers (e.g., imaging biomarkers, blood biomarkers, cerebrospinal fluid biomarkers).

## Data extraction

One reviewer (WQ) designed the data charting form, which was verified by other authors. The data charting form will be designed using the Critical Appraisal and Data Extraction for Systematic Reviews of Prediction Modelling Studies (CHARMS) checklist [31]. Predefined data charts will be independently extracted and recorded in Excel by one reviewer

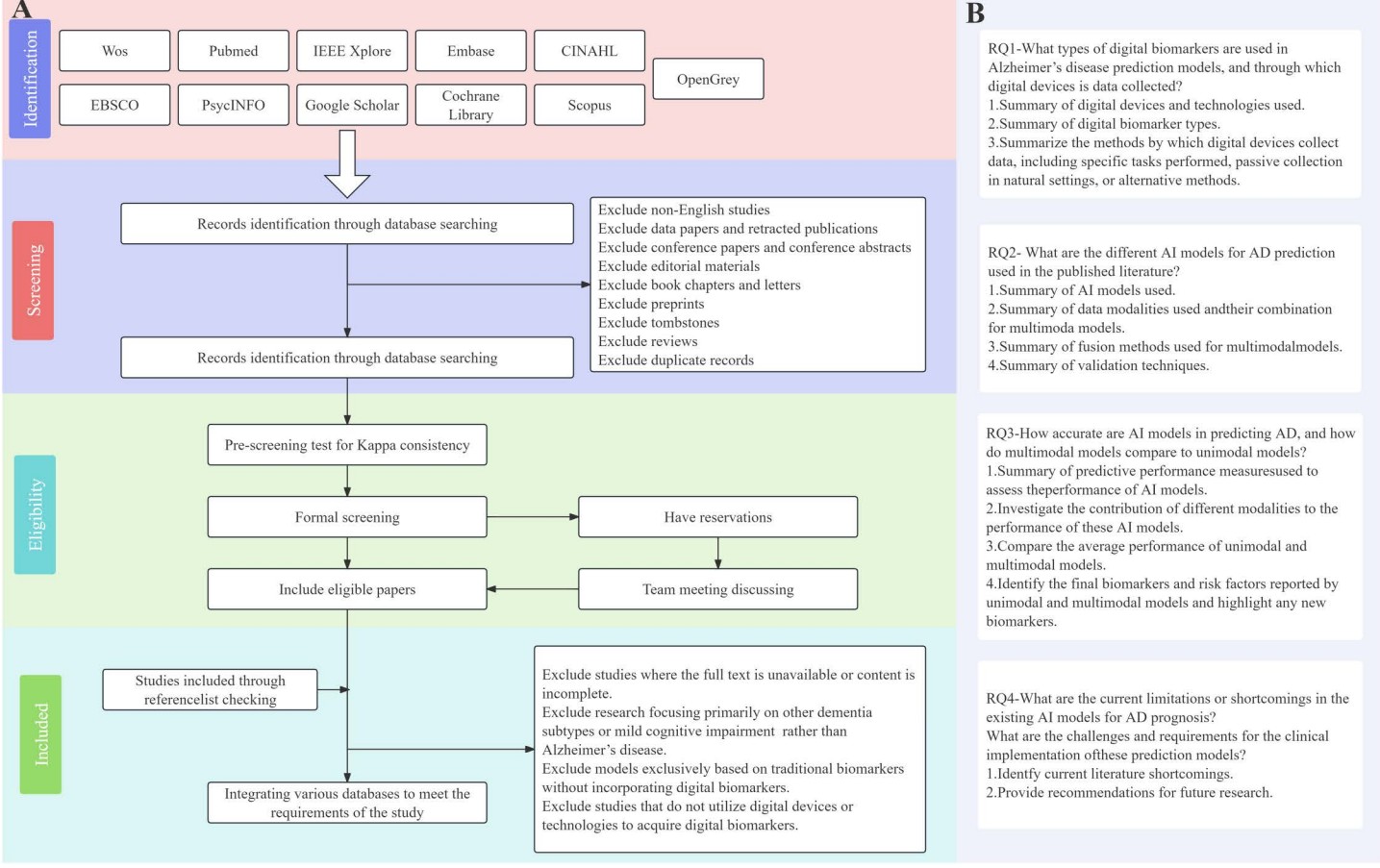

**Fig 2. Study inclusion and exclusion Flowchart.**

(WQ), with all extracted data verified by a second reviewer (CD). All data items will be extracted at the individual study level, and the reporting for this scoping review will adhere to the PRISMA guidelines and its relevant extensions. Once the data charting form is completed and the evidence clarified, both reviewers (WQ and CD) will draft a narrative summary, presenting the summarized results in tabular or interactive visualization formats. The CHARMS data extraction method published by the Cochrane Prognosis Methods Group is provided in Supplementary S3 Table online. We have made appropriate adjustments, with additional requirements to extract information on reproducibility, reporting standards, and multimodal fusion strategies. As the study progresses, we will update the content of the data extraction form as needed to align with CHARMS data extraction benchmarks. The data extraction form is presented in Supplementary S4 Table online.

The PROBAST tool comprises 20 questions across four domains (participants, predictors, outcomes, and analysis) [25]. Two independent reviewers will evaluate the risk of bias (RoB) and the applicability of each study using these questions. If a study shows no shortcomings across all domains, the RoB is rated as "low," and applicability as "low concern"; if there is a deficiency in at least one domain, the RoB is rated as "high," and applicability as "high concern." If there is insufficient information to clearly assess the risk of bias, the study is categorized as having "unclear risk of bias." RoB and applicability assessment results will be tabulated according to the PROBAST guidance and interpretation document (see Supplementary S5 Table online).

TRIPOD will be used to assess the transparency and completeness of reporting in prediction modeling studies [32]. TRIPOD consists of 22 items and their subitems, each of which will be independently checked by two reviewers and marked as "yes," "no," or "not applicable." For calibration, both tools will undergo pilot testing on a sample of five selected articles. If the inter-rater Kappa value surpasses 0.8, the tools will then be applied to the remaining studies. Should this threshold not be reached, discrepancies will be analyzed, followed by recalibration to ensure consensus.

## Data synthesis

Screening Results: We will use a PRISMA flowchart to display the results of literature search and screening, documenting the number of records identified in each database, the records meeting eligibility criteria, and the final number of studies included in the scoping review.

Study Interpretation: Relevant information will be analyzed and summarized based on the PRISMA-ScR checklist. A narrative report will consolidate the overall findings related to the research questions, and tables will be used to present detailed information for each included study.

Study Assessment: A PROBAST table will be created to display the results of the risk of biasand applicability assessments. In the TRIPOD assessment, the percentage of compliance with all relevant items will be calculated for each study, with specific reports on RoB and TRIPOD evaluation results.

Data Sources: After extracting the data types for the study, the models are initially categorized into unimodal and multimodal models based on the nature of the modalities. For unimodal models, further classification is performed based on the specific data sources captured (e.g., variations in the type of wearable devices and their placement locations lead to differences in the types of data collected). These models are expected to include gait models, manual models, physiological signal models, speech models, and possibly eye-tracking models. For multimodal models that involve multiple data types or integrate omics data, scale information, and other content, we will further discuss the fusion methods of the root modalities (early fusion, mid-fusion, late fusion, and hybrid fusion). Additionally, we will present the performance distribution differences between unimodal and multimodal models.

Regarding the performance of models built with different types of data, we will first present the performance distribution differences between unimodal and multimodal models, covering metrics such as accuracy, specificity, sensitivity, and AUC. For studies where performance data is not reported, only the publicly available performance metrics will be compared. In each study, we will compare the performance differences between the models with the best reported performance. In multimodal studies, unimodal and multimodal models are typically constructed and compared, with specific performance metrics provided to highlight the differences in performance between these models. The best data type combinations will be summarized.

Additionally, we will conduct a comparative analysis of studies based on different data types, such as comparing the performance differences between multimodal and unimodal gait models. These differences will provide further insights into each type of model.

Furthermore, regarding the impact of data collection tasks and devices on performance, we will compile the combinations of collection tasks and devices based on the classification of each model type. This will allow us to compare which devices provide the best performance for tasks such as simple gait tasks, under a consistent collection task. We will also examine the performance differences for similar wearable devices across different gait tasks (e.g., single-task versus dual-task gait), thus gaining insights into how diverse data collection methods influence model performance.

Model Comparison: Models will be categorized based on their predictive outcomes or classification tasks. All extracted data regarding model development or validation will be presented in tabular form. For studies that developed multiple models, only the highest-performing model will be extracted for comparative performance analysis and summary.

Discussion Module Preparation: The results will be interpreted in line with the study objectives and goals, identifying deficiencies in the current literature regarding reporting standards and development methods for AD predictive models. All results are expected to be officially presented in June 2025.

## Discussion

The scoping review will regularly examine the latest literature, aiming to capture relevant studies as comprehensively as possible. Research findings will be disseminated through manuscript submission to relevant journals. Any revisions to the study protocol will be publicly available on the OSF platform.

Alzheimer's disease is currently the most prevalent neurodegenerative disorder, and with the ongoing global population aging, it is expected to become a significant public health challenge in the future. As the demand for non-invasive, cost-effective, and personalized diagnostic methods increases, the use of wearable devices combined with artificial intelligence models is becoming increasingly important [33]. Moreover, with the introduction of the FDA's SaMD initiative, the integration of these two technologies is becoming more feasible and moving toward commercialization [34]. While predictive models using wearable devices and AI algorithms have gained popularity in recent years, their architectures and performance remain unclear. This scoping review aims to address this gap and provide critical insights into these models in the given context.

In summary, the contribution of this protocol lies not only in summarizing existing research but also in advancing the application of AI and wearable devices in Alzheimer's disease (AD) diagnosis and management. This work fills gaps in current research and provides guidance for future studies and clinical practice. Firstly, the planned review study comparing the performance of different algorithms in AD diagnosis will offer clear directions for future research. By identifying which algorithms are more effective for specific tasks, researchers can focus on optimizing these algorithms, thereby improving diagnostic accuracy and predictive capabilities. For models facing performance bottlenecks, further research could concentrate on improving model architectures or adopting new algorithmic strategies. This would not only enhance algorithm efficiency but also potentially expand their application in diagnosing other related diseases. For instance, to address the limitations of deep learning algorithms in data-scarce scenarios, future studies could explore the use of transfer learning or data augmentation techniques to improve performance on small sample datasets.

Secondly, this protocol emphasizes research on pairing wearable devices with clinical tasks, highlighting the strong correlation between devices and tasks. Future studies, based on the summary provided by the protocol, can further explore which types of tasks are best suited for pairing with specific devices, thus increasing the efficiency of clinical applications. This finding will provide valuable guidance to device developers, helping them create more precise and efficient diagnostic tools. Additionally, issues with missing data during the data collection process offer direction for improving data acquisition technologies. Future studies could explore new data collection methods and technologies, such as the integration of augmented reality (AR) or virtual reality (VR) with wearable devices, to capture richer physiological and behavioral data, thereby providing more data for early AD diagnosis.

Furthermore, by comparing models based on different collection tasks, devices, modalities, and data types, we can gain a deeper understanding of how diverse data collection methods influence model performance. Different collection tasks and devices directly impact data quality. Through the study of these diverse data collection methods, we can uncover how various factors interact synergistically, enhancing the model's performance across different application scenarios. This is particularly relevant in the fields of healthcare and health management, providing both theoretical and practical support for optimizing existing models and developing accurate diagnostic tools.

Moreover, regarding multimodal data integration, this research will emphasize the impact of application differences and performance distributions across diverse datasets, encouraging future studies to focus on effectively combining multi-source data. For example, future studies could explore how to deeply fuse different types of data (such as behavioral data, electroencephalogram (EEG) data, and physiological signals) to further enhance diagnostic accuracy. In clinical applications, these findings will help clinicians understand the relationship between different devices, algorithms, and tasks, thereby enabling more personalized treatment plans in clinical practice.

Finally, the initiative of publicly releasing the code will serve as a bridge between the academic and clinical communities. By increasing transparency and reproducibility, researchers and clinicians will be able to more easily validate results, optimize algorithms, and customize development based on varying clinical needs. This will facilitate the deeper application of AI in healthcare, ultimately improving the diagnostic efficiency and accuracy of diseases such as AD.

Overall, conducting the further review according to this protocol not only provides strong theoretical support for technological advancements in AD diagnosis but also offers feasible pathways for optimizing clinical applications and integrating multimodal data. These findings will directly drive future research directions and help clinical practice achieve greater breakthroughs in early disease detection and personalized treatment.

## Supporting information

**S1 Table.  (PRISMA-ScR) Checklist.**
(DOCX)

**S2 Table.  Search strategies.**
(DOCX)

**S3 Table.  CHARMS 2014.**
(DOCX)

**S4 Table.  Charting data form.**
(DOCX)

**S5 Table.  PROBAST form.**
(DOCX)

**S1 Prisma.  P-checklist (1).**
(DOCX)

## Patient and public involvement

This study does not involve patients or the public.

## Author contributions

**Conceptualization:** Lizzy M. M. Boots, Shihua Cao.

**Data curation:** Yankai Shi.

**Formal analysis:** Yankai Shi.

**Investigation:** Shiying Shen.

**Methodology:** Yankai Shi, Shiying Shen, Lizzy M. M. Boots, Qian Xu.

**Resources:** Bingsheng Wang, Qian Xu.

**Software:** Bingsheng Wang.

**Supervision:** Bingsheng Wang, Shuai Zhang, Junling Kang, Xiaodong Lu, Lizzy M. M. Boots, Qian Xu, Shihua Cao.

**Validation:** Shuai Zhang, Junling Kang, Xiaodong Lu.

**Visualization:** Wenhao Qi, Xiaodong Lu.

**Writing – original draft:** Wenhao Qi, Guowei Jiang.

**Writing – review & editing:** Wenhao Qi, Guowei Jiang.

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
