## [Decision Letter · Decision Letter 0]

11 Mar 2025

PLOS ONE

Dear Dr. Cao,

Thank you for submitting your manuscript to PLOS ONE. After careful consideration, we feel that it has merit but does not fully meet PLOS ONE’s publication criteria as it currently stands. Therefore, we invite you to submit a revised version of the manuscript that addresses the points raised during the review process.

The manuscript presents a study protocol examining the role of wearable devices and artificial intelligence (AI) in the management and prevention of Alzheimer's disease (AD). While it addresses an important and emerging research area, several aspects require refinement to enhance clarity, methodological rigor, and theoretical grounding.

<h4 data-end="422" data-start="385">**Key Areas for Improvement:** </h4> 

**Abstract Clarity**The abstract is overly redundant and structured more like an introduction rather than a concise summary. A complete rewrite is recommended to succinctly present the study objectives, methodology, and anticipated contributions.**Methodology Enhancements**The definition of gray literature should be clarified with examples to improve the transparency of the search strategy.The study employs the **PROBAST** and **TRIPOD** tools, but their specific roles (e.g., PROBAST for risk of bias assessment, TRIPOD for reporting completeness) should be explicitly stated within the methods section.There is an inconsistency between the databases listed in the methodology and the supplementary material. While **Google Scholar, Scopus, and Cochrane Library** are mentioned in the main text, the supplementary files do not include corresponding search strategies. Example search strings for these databases should be added for reproducibility.**Scope and Contribution**The study explores a relevant intersection of **wearable technology, AI, and AD management** , but its unique contribution is unclear. The introduction should explicitly state how this review addresses gaps in the existing literature.The protocol promises to synthesize prior research but does not specify how its findings will **advance clinical practice or guide future research** . A stronger discussion on this aspect is needed.The manuscript does not sufficiently address the theoretical models underpinning AI applications in AD prediction. A review of existing frameworks, such as **machine learning principles adapted for healthcare** , would strengthen the study's foundation.

**Data Handling Considerations**The review will cover diverse data sources, including **gait analysis and EEG data** , yet there is limited discussion on how multimodal data will be handled. More details are needed on how different data collection methods may influence model performance.

**Lack of Novelty**While the study protocol is well-designed, it does not introduce new findings. The manuscript should highlight its **newly added value** to distinguish it from existing reviews in the field

We look forward to receiving your revised manuscript.

Kind regards,

Muhammad Shahzad Aslam, Ph.D.,M.Phil., Pharm-D

Academic Editor

PLOS ONE

Journal Requirements:

2. Thank you for stating the following financial disclosure: [Zhejiang Province Traditional Chinese Medicine, Science, and Technology Project (2023ZF134) supported by SC, Zhejiang Provincial Medical and Health Science and Technology Program Project (2022KY1052) supported by SC,First-Class Course of Zhejiang Province (2022-1133) supported by SC,Basic Public Welfare Research Project/Joint Fund Project of Zhejiang Province: Research on accurate localization of gait disorder brain region in Parkinson's disease based on pfMRI data set and hierarchical Bayesian model(LBY23H200002), supported by XL.]. 

Reviewers' comments:

Reviewer's Responses to Questions

**Comments to the Author**

1. Does the manuscript provide a valid rationale for the proposed study, with clearly identified and justified research questions?

Reviewer #1: Yes

Reviewer #2: Yes

Reviewer #3: Yes

Reviewer #4: Yes

2. Is the protocol technically sound and planned in a manner that will lead to a meaningful outcome and allow testing the stated hypotheses?

Reviewer #1: Yes

Reviewer #2: Yes

Reviewer #3: Yes

Reviewer #4: Yes

3. Is the methodology feasible and described in sufficient detail to allow the work to be replicable?

Reviewer #1: Yes

Reviewer #2: Yes

Reviewer #3: Yes

Reviewer #4: Yes

4. Have the authors described where all data underlying the findings will be made available when the study is complete?

Reviewer #1: Yes

Reviewer #2: Yes

Reviewer #3: Yes

Reviewer #4: Yes

5. Is the manuscript presented in an intelligible fashion and written in standard English?

Reviewer #1: Yes

Reviewer #2: Yes

Reviewer #3: Yes

Reviewer #4: Yes

You may also provide optional suggestions and comments to authors that they might find helpful in planning their study.

Reviewer #1: My concerns are as follows:

1. The abstract is overly redundant and appears to be written as an introduction; a complete rewrite of the abstract is recommended.

2. The paper explores a relatively emerging intersection of wearable devices and AI in Alzheimer's disease (AD). However, the existing literature on AI in healthcare, including Alzheimer's, has been growing, and the manuscript does not highlight its unique contribution clearly in the introduction. It would benefit from a clearer statement on how the review will fill existing gaps.

3. The protocol promises to synthesize a significant amount of existing research, but it remains unclear what specific contributions it will make beyond summarizing previous findings. It would be useful to further emphasize how the findings will directly influence future research directions or clinical practices. Some workss are worth mentioning: (1)DOI:10.62762/CJIF.2024.876830 (2)DOI:10.62762/TIS.2024.680959

4. The manuscript discusses the types of data (e.g., gait, EEG) that will be covered, but there is limited discussion on how diverse data collection methods could influence the model's performance. It should address how the variety of data sources will be handled, especially multimodal data.

5. What are the newly-added values?

6. The manuscript lacks a detailed discussion on the theoretical models that underpin the use of AI in AD prediction. A more comprehensive review of existing theories or frameworks (such as machine learning principles and their adaptation for healthcare applications) would strengthen the theoretical grounding.

Reviewer #2: This is a very interesting manuscript. The use of wearable devices and the use of artificial intelligence as an intervention in the management and prevention of Alzheimer's disease is welcomed.

Reviewer #3: Although the protocol described in this manuscript itself is well designed, it does not provide any new findings. The authors should perform a scoping review based on their own protocol and publish the results.

Reviewer #4: First off, congrats on putting together this study protocol. Now, as I was reading through, a couple of things popped up that you might want to think about a bit more:

Methods

One thing to consider clarifying is regarding the "gray literature" you plan to retrieve. Could you give a few examples of what you consider gray literature in this context? This would make your search strategy even more transparent.

You mention using the PROBAST and TRIPOD tools. It might be helpful to briefly mention what each of these tools helps you assess (e.g., risk of bias for PROBAST, reporting completeness for TRIPOD) directly in the methods section rather than just citing them.

Regarding the databases mentioned in the methodology versus the supplementary material, it's important to note that the main "Search strategy" section explicitly lists Google Scholar, Scopus, and Cochrane Library (along with other databases) as sources that will be searched. However, in Supplementary file, which provides example search strategies for different databases, a specific search strategy for Google Scholar, Scopus, or Cochrane Library is not explicitly shown. For transparency and reproducibility, it would be beneficial for the authors to include example search strings for these databases as well in the supplementary material.

Overall, I must say that this is a very well-put-together study protocol, and I genuinely enjoyed reading it. Addressing these minor points will enhance the clarity, transparency, and rigor of your protocol. To be honest, I see significant potential for publication here if these enhancements are made. Your detailed methodology and focus on a crucial area of research make this a valuable contribution. I wish you the best of luck with your scoping review.

**Do you want your identity to be public for this peer review?** For information about this choice, including consent withdrawal, please see our Privacy Policy

Reviewer #1: No

Reviewer #2: **Yes: ** OLAYINKA ADEBAJO

Reviewer #3: No

Reviewer #4: No

---

## [Author Response · Author response to Decision Letter 1]

31 Mar 2025

March 30, 2025

Dear Editor,

Dear Reviewers,

Dear PLOS One Editorial Office,

We have submitted the revised manuscript entitled "Integration of wearable devices and artificial intelligence in Alzheimer's disease: a scoping review protocol" (Submission ID: PONE-D-25-03879R1). Additionally, we have provided a version of the manuscript with tracked changes for review.

We sincerely appreciate the efforts of the editorial team and the constructive feedback provided by the reviewers. The suggestions were extremely valuable and played a crucial role in improving the quality of our manuscript. These recommendations have been essential in guiding our research. We have made revisions in response to each comment from the four reviewers, and we have reorganized the content in the attached document accordingly.

1.Given the increasing popularity of integrating AI with wearable devices, it has become more important than ever to write a review on Alzheimer’s disease in this context.

2.To our knowledge, this manuscript represents the first scoping review specifically focusing on AI models for wearable devices in Alzheimer’s disease.

3.We anticipate that executing this protocol will yield meaningful insights that will contribute to clinical applications, such as model validation and performance distribution.

We hope that this revised manuscript meets the expectations of both the journal and the reviewers, ultimately allowing it to reach a broader audience. Our goal is to facilitate knowledge transfer in this field and make a significant contribution to its development. Therefore, we are hopeful that the revised manuscript will be accepted for publication in PLOS One.

Corresponding Author: Shihua Cao

Email: csh@hznu.edu.cn

Key Laboratory of Cognitive Disorder Assessment Technology

Hangzhou, China

Please ensure that your manuscript meets PLOS ONE's style requirements, including those for file naming. The PLOS ONE style templates can be found athttps://journals.plos.org/plosone/s/file?id=wjVg/PLOSOne_formatting_sample_main_body.pdf andhttps://journals.plos.org/plosone/s/file?id=ba62/PLOSOne_formatting_sample_title_authors_affiliations.pdf

Response to Editor's comment 1:

We appreciate the editor's comments regarding the manuscript formatting. We have made the necessary adjustments in accordance with the journal's formatting guidelines. Specifically, we have:Added table titles following the references section. Modified the font sizes for section, subsection, and sub-subsection headings. Bolded the prefix of the figure titles. Ensured that all headings are written in sentence case, as per the journal’s requirements.

2. Thank you for stating the following financial disclosure: [Zhejiang Province Traditional Chinese Medicine, Science, and Technology Project (2023ZF134) supported by SC, Zhejiang Provincial Medical and Health Science and Technology Program Project (2022KY1052) supported by SC,First-Class Course of Zhejiang Province (2022-1133) supported by SC,Basic Public Welfare Research Project/Joint Fund Project of Zhejiang Province: Research on accurate localization of gait disorder brain region in Parkinson's disease based on pfMRI data set and hierarchical Bayesian model(LBY23H200002), supported by XL.]. 

Please state what role the funders took in the study.  If the funders had no role, please state: ""The funders had no role in study design, data collection and analysis, decision to publish, or preparation of the manuscript.""If this statement is not correct you must amend it as needed.Please include this amended Role of Funder statement in your cover letter; we will change the online submission form on your behalf.

Response to Editor's comment 2:

We appreciate the editor's suggestion regarding the funding statement. In response, we have added the following statement in the Acknowledgments section: "The funders had no role in study design, data collection and analysis, decision to publish, or preparation of the manuscript." We also included a statement about the role of funders in the second-to-last paragraph of the cover letter: "The funders had no role in study design, data collection and analysis, decision to publish, or preparation of the manuscript."

3.When completing the data availability statement of the submission form, you indicated that you will make your data available on acceptance. We strongly recommend all authors decide on a data sharing plan before acceptance, as the process can be lengthy and hold up publication timelines. Please note that, though access restrictions are acceptable now, your entire data will need to be made freely accessible if your manuscript is accepted for publication. This policy applies to all data except where public deposition would breach compliance with the protocol approved by your research ethics board. If you are unable to adhere to our open data policy, please kindly revise your statement to explain your reasoning and we will seek the editor's input on an exemption. Please be assured that, once you have provided your new statement, the assessment of your exemption will not hold up the peer review process.

Response to Editor's comment 3:

We appreciate the editor's reminder regarding the data statement. All authors have agreed to the statement that all data in the manuscript will be made freely accessible.

Response to Editor's comment 4:

We appreciate the editor's reminder. The corresponding author has now verified and linked their ORCID. Finally, we would like to express our sincere gratitude to the editor for the meticulous review, which has helped us refine the formatting details of the manuscript and assisted us in adhering to the journal's required declarations.

Reviewer #1: My concerns are as follows:

1. The abstract is overly redundant and appears to be written as an introduction; a complete rewrite of the abstract is recommended.

Response to Reviewer #1's comment 1:

We appreciate your detailed comments on the abstract. Indeed, our original abstract was too lengthy, and as per your suggestion, we have rewritten it. This revision has significantly improved the clarity of the abstract. Thank you once again for your constructive feedback.

Modified content: The incidence of Alzheimer's disease continues to rise, and predictive models combining artificial intelligence (AI) with wearable devices offer a new approach for its detection and diagnosis. Given the rapid development of these emerging technologies, a comprehensive review is necessary to supplement the evidence in this field. This review aims to systematically evaluate AI-based predictive models using wearable devices, with a focus on their measurement outcomes and model development processes. Following the Arksey and O'Malley framework and incorporating PRISMA and its extension guidelines, this study will search multiple databases, including Web of Science, Cochrane Library, and PubMed, covering relevant gray literature. The quality of the included studies will be rigorously assessed using the Prediction Model Risk of Bias Assessment Tool (PROBAST) and Transparent Reporting of a Multivariable Prediction Model for Individual Prognosis or Diagnosis (TRIPOD) checklists. Two independent reviewers will conduct title and abstract screening, retrieve and assess full-text evidence sources, and extract data. The results will be narratively synthesized and presented in tables and figures. This review is expected to provide systematic evidence to support predictive models combining wearable devices and AI, contributing to the advancement of this field.

(Lines 26-42)

2. The paper explores a relatively emerging intersection of wearable devices and AI in Alzheimer's disease (AD). However, the existing literature on AI in healthcare, including Alzheimer's, has been growing, and the manuscript does not highlight its unique contribution clearly in the introduction. It would benefit from a clearer statement on how the review will fill existing gaps.

Response to Reviewer #1's comment 2:

We greatly appreciate your detailed and constructive feedback on our research. The previous version of the manuscript indeed lacked sufficient emphasis on how our work addresses existing research gaps. In response to your suggestion, we have revised the introduction to highlight the contributions of our study and how it fills these gaps. This addition effectively clarifies our research's unique contribution to the field.

(1)First, we have outlined the limitations of existing studies and explained how their focus differs from ours.

Modified content: The potential of these technologies is being explored; however, only two systematic reviews [19, 20] have addressed the application of wearable devices and AI models in AD. These reviews cover a broader range of topics and primarily focus on studies related to clinical biomarker data, such as radiomics. They do not provide detailed reports on the deployment strategies of wearable devices or the specifics of model construction. Additionally, these reviews fail to adequately demonstrate the feasibility of integrating wearable devices with AI technologies.

(Lines 96-102)

(2)We have also added a section outlining the contributions of our study and the gaps it addresses.

Modified content: The unique contribution of this protocol lies in its development of a systematic evaluation and comprehensive review of AI prediction models based on wearable devices. By providing a comprehensive review of existing research, it addresses the gap in the current literature concerning the integration of these two technologies. Specifically, we aim to report on several key dimensions, including the selection of wearable devices, data collection paradigms, feature selection strategies, model construction and training, model validation, and model transparency. In addition to analyzing the applications of various wearable devices, we explore how advanced AI techniques can enhance their diagnostic accuracy and monitoring capabilities. This approach offers new perspectives for the early diagnosis of AD, prediction of disease progression, and personalized healthcare.

(Lines 111-121)

3. The protocol promises to synthesize a significant amount of existing research, but it remains unclear what specific contributions it will make beyond summarizing previous findings. It would be useful to further emphasize how the findings will directly influence future research directions or clinical practices. Some workss are worth mentioning: (1)DOI:10.62762/CJIF.2024.876830 (2)DOI:10.62762/TIS.2024.680959

Response to Reviewer #1's comment 3:

(1)We greatly appreciate your suggestion regarding the specific contributions of our manuscript. Due to our previous unclear statement, this section was insufficient. In response to your feedback, we have made the necessary revisions. We have expanded the discussion section to include the contributions of the content we have collected to clinical practice and future research. Thank you once again for this specific and valuable suggestion, which has greatly improved the quality of our manuscript.

Modified content: Specifically, we anticipate providing a series of clinically meaningful insights. First, the summary of wearable devices and data collection paradigms will highlight the practical applications and limitations of current technologies in AD diagnosis. For instance, the pairing of different types of devices with various tasks will guide the design of clinical diversity, while the identification of causes for data gaps during the collection process will help recognize potential issues with existing data acquisition technologies. Furthermore, these insights can provide direction for future improvements in device design and data collection methods. The analysis of the quantity, types, and performance distributions of the algorithms currently in use will provide a clear representation of the field's status and development trends. By examining the application of different algorithm models and their performance across various datasets, we can identify which algorithms perform best in AD diagnosis and prediction tasks, as well as which models still face performance bottlenecks. This study will also reveal the applicability of various models in different clinical scenarios (e.g., model training costs, device portability, and data acquisition challenges), helping to identify which models are best suited for early diagnosis and disease progression tracking. Although we do not anticipate directly comparing the performance of unimodal and multimodal models, but rather presenting the performance distribution (due to the objective influences of differences in data acquisition devices, tasks, feature types, and algorithm architectures), this approach will still provide valuable insights into the combination of multimodal data. Furthermore, an analysis of external validation, calibration, and related aspects will help assess the generalizability and reliability of these models across different datasets and real-world clinical environments. The public release of code is essential for promoting transparency and reproducibility within the field. In summary, this scoping review will not only highlight the strengths of current technologies in the field of AD but will also assist researchers and clinicians in understanding the challenges that may arise in practical applications.

(Lines 398-427)

(2)The two studies you mentioned regarding EEG data processing are indeed highly relevant to the design of our protocol. In response, we have added these two studies in the introduction to enrich the theoretical background on the progress between machine learning algorithms and data processing. Your suggestion has greatly helped improve the completeness of our manuscript. Thank you once again for this constructive feedback.

Modified content: Similarly, graph neural networks play a crucial role in the spatiotemporal processing of EEG data by simultaneously considering the dynamic changes in the time series and the spatial relationships between different brain regions[17,18].

(Lines 92-95)

4. The manuscript discusses the types of data (e.g., gait, EEG) that will be covered, but there is limited discussion on how diverse data collection methods could influence the model's performance. It should address how the variety of data sources will be handled, especially multimodal data.

Response to Reviewer #1's comment 4:

We greatly appreciate your specific suggestions regarding the data processing of different models in our manuscript. While we had considered multimodal and different data collection, it was not sufficiently presented in the original manuscript. In response, we have added a more detailed description of the data processing in the Data Synthesis section. This revision strengthens our research methodology and enriches the content of our st

---

## [Decision Letter · Decision Letter 1]

29 May 2025

Dear Dr. Cao,

Thank you for submitting your manuscript to PLOS ONE. After careful consideration, we feel that it has merit but does not fully meet PLOS ONE’s publication criteria as it currently stands. Therefore, we invite you to submit a revised version of the manuscript that addresses the points raised during the review process.

We look forward to receiving your revised manuscript.

Kind regards,

Marcello Moccia

Academic Editor

PLOS ONE

Additional Editor Comments:

Please, carefully consider new comments and also improve responses to Reviewer 1.

Reviewers' comments:

Reviewer's Responses to Questions

**Comments to the Author**

1. Does the manuscript provide a valid rationale for the proposed study, with clearly identified and justified research questions?

Reviewer #1: No

Reviewer #2: Yes

Reviewer #3: Yes

2. Is the protocol technically sound and planned in a manner that will lead to a meaningful outcome and allow testing the stated hypotheses?

Reviewer #1: No

Reviewer #2: Yes

Reviewer #3: Yes

3. Is the methodology feasible and described in sufficient detail to allow the work to be replicable?

Reviewer #1: No

Reviewer #2: Yes

Reviewer #3: Yes

4. Have the authors described where all data underlying the findings will be made available when the study is complete?

Reviewer #1: No

Reviewer #2: Yes

Reviewer #3: Yes

5. Is the manuscript presented in an intelligible fashion and written in standard English?

Reviewer #1: Yes

Reviewer #2: Yes

Reviewer #3: Yes

You may also provide optional suggestions and comments to authors that they might find helpful in planning their study.

Reviewer #1: The author did not address my previous concerns and therefore the current version is not suitable for publication.

Reviewer #2: This is such an insightful article , although not novel, as there are similar articles that have explored the use of AI and wearable devices in managing alzheimer's disease. This article adds to the body of knowledge.

Reviewer #3: I suppose that the revised manuscript has the content and quality to be published as a study protocol paper.

**Do you want your identity to be public for this peer review?** For information about this choice, including consent withdrawal, please see our Privacy Policy

Reviewer #1: No

Reviewer #2: **Yes: ** OLAYINKA ADEBAJO

Reviewer #3: **Yes: ** Ryuichiro ARAKI

---

## [Author Response · Author response to Decision Letter 2]

12 Jun 2025

Dear Editor and Reviewers,

We have made extensive revisions to the manuscript, and the changes are presented in a table format for clarity. However, due to system limitations, we are unable to fully display these changes here. Therefore, following the journal's suggestion, we have submitted the revised manuscript along with the response to the reviewers' comments as an attachment, named "Round 2. Response to the reviewers' comments." Please ensure that this attachment is not confused with the one from the first round of revisions.

Best wishes for your work.

Shihua Cao

Hangzhou normal university

---

## [Editor Report · Decision Letter 2]

12 Aug 2025

Integration of wearable devices and artificial intelligence in alzheimer's disease: a scoping review protocol

PONE-D-25-03879R2

Dear Dr. Cao,

We’re pleased to inform you that your manuscript has been judged scientifically suitable for publication and will be formally accepted for publication once it meets all outstanding technical requirements.

Kind regards,

Marcello Moccia

Academic Editor

PLOS ONE

Additional Editor Comments (optional):

Thank you for addressing the comments and, again, sorry if the peer review process took longer than we expected.
---

## [Editor Report · Acceptance letter]

PONE-D-25-03879R2

PLOS ONE

Dear Dr. Cao,

I'm pleased to inform you that your manuscript has been deemed suitable for publication in PLOS ONE. Congratulations! Your manuscript is now being handed over to our production team.

Kind regards,

on behalf of

Dr. Marcello Moccia

Academic Editor

PLOS ONE